# Attainment Targets for Protein Intake Using Standardised, Concentrated and Individualised Neonatal Parenteral Nutrition Regimens

**DOI:** 10.3390/nu11092167

**Published:** 2019-09-10

**Authors:** Colin Morgan, Maw Tan

**Affiliations:** 1Liverpool Women’s Hospital, Liverpool L8 7SS, UK; 2Alder Hey Children’s Hospital, Liverpool L14 5AB, UK

**Keywords:** neonatal, parenteral nutrition, preterm, protein, amino acid, standardisation, target attainment, growth, electrolyte, mineral, pharmacy, safety

## Abstract

Neonatal parenteral nutrition (NPN) regimens that are individualised (iNPN) or standardised concentrated NPN (scNPN) are both currently used in preterm clinical practice. Two recent trials (one iNPN and one scNPN) each compared standard (control) and high (intervention) parenteral protein and energy dosage regimens and provided data about actual protein intake. We hypothesised that scNPN regimens would achieve a higher percentage of the target parenteral protein intake than their corresponding iNPN regimens. We calculated the daily individual target parenteral protein intake and used the daily parenteral protein intake to calculate the target attainment for protein intake in each infant for the two control (iNPN: *n* = 59, scNPN: *n* = 76) and two intervention (iNPN: *n* = 65; scNPN: *n* = 74) groups. The median (IQR) target attainment of high-dose protein was 75% (66–85) versus 94% (87–97) on days 1–15 for iNPN and scNPN regimens respectively (*p* < 0.01). The median (IQR) target attainment of standard dose protein was 77% (67–85) versus 94% (91–96) on days 1–15 for iNPN and scNPN regimens, respectively (*p* < 0.01). This was associated with improved weight gain (*p* = 0.050; control groups only) and head growth (*p* < 0.001; intervention groups only). scNPN regimens have better target attainment for parenteral protein intakes than iNPN regimens.

## 1. Introduction

Neonatal parenteral nutrition (NPN) is an essential element of preterm care. The potential for early nutritional deficits in very preterm infants has been long understood [1,2] and previous nutritional recommendations have been a contributing factor [3]. Many studies have suggested such deficits are preventable [4,5,6,7,8] with benefits for growth and potentially other outcomes. The provision of NPN services is highly complex, requiring high-quality pharmacy aseptic manufacturing services, robust and flexible delivery systems to the health provider, sound prescribing/dispensing protocols at local pharmacy level and detailed clinical and administration guidelines in the neonatal service. Much of the variation in NPN macronutrient intake (glucose, protein or lipid intake) results from differences in nutritional policy [9], but the complexity of the supply chain has great scope for introducing errors [10,11] and unintended variation. Over the last decade, national survey and audit data have repeatedly demonstrated large variation in neonatal PN provision [12,13,14,15,16,17,18]. Some of the differences in nutritional intake reflect nutritional policy (i.e., intended variation). However, differences between patients in the same service and inconsistencies in NPN supply chains between neonatal services indicate much of the variation in actual nutrient intake was unintended, particularly in the UK [15,16,17,18].

The conventional NPN strategy has been based on individualised prescription and formulation to address the rapidly changing and variable fluid and electrolyte needs characteristic of the very preterm infant. This has the potential to subvert early nutritional strategy particularly with inexperienced neonatal PN prescribers [16,17]. Computer-aided prescribing can help [18] and improve protein and energy intake [19,20,21]. Individualised NPN prescription and formulation does create major practical challenges for supply chains, thereby introducing unintended variation in actual NPN delivery to the patient. Although individualised PN prescription is flexible, the manufactured individualised PN bag is not. It takes time to manufacture and requires resources which are not consistently available 24 h, 7 days a week. Quality assurance of manufactured individualised bags is much more limited than for those that can be batch tested. Individualised NPN does not allow rapid responses to changes in fluid and electrolyte requirements after the PN bag had been prescribed, potentially compromising nutrient intake or increasing wastage.

Standardised versus individualised neonatal PN has been reviewed [22,23], including systematic review [24]. Recent NPN guidance now recommends that NPN should be standardised where possible, reserving iNPN for the more complex cases [24]. A target of 80% standardised NPN use in preterm infants has been proposed [25]. While some evidence suggests iNPN may be beneficial [20,26,27], increasing evidence indicates that most infants can be managed on a standardised PN formulation [8,28,29,30,31,32,33,34,35] One of the important ways in which standardisation helps improve nutrition is allowing PN to be started immediately after birth. The provision of early parenteral protein (as amino acids) is particularly important for early nitrogen retention, reducing early nutritional deficits and improving growth [8,35,36,37]. This also minimises interpatient variation in nutritional management, resulting from lack of PN services over weekends [38]. There are also other benefits from standardization, including quality assurance during manufacture, simplification of the supply chain and safer prescribing/administration [24,25].

While unintended variation due to NPN prescribing practice has been investigated, little attention has been given to the process of PN administration, which may vary considerably from the original prescription given the time-lag between laboratory results being received and NPN manufactured, dispensed and connected to the patient. Even then, complex and rapidly changing fluid and electrolyte requirements may have altered the preterm infant’s requirements. Drug infusions and fluid restriction can also limit the volume of fluid available for aqueous PN (aqNPN). This contains all protein (as amino acids) and other water-soluble nutrients and maintenance electrolytes. This effect can be reduced by concentrating the aqueous PN into a smaller volume and making up any additional fluid requirement with a supplementary infusion (for example, 10% glucose) [8,39,40,41] It is then the 10% glucose infusion that is titrated up and down if total fluid requirements change or additional infusions are added or altered. This “protects” the aqNPN infusion and ensures continuity of nutrient intake. The degree of concentration is limited by the stability of the aqNPN, so only standardised solutions that have undergone high quality assurance associated with batch testing should be used. There are now a range of different aqNPN bags available at different concentrations. This approach assumes the neonatal lipid infusion is separately administered.

Standardising and concentrating neonatal PN (scNPN) has the potential to address the problems of suboptimal nutrient administration but randomised controlled trial (RCT) comparisons of scNPN and iNPN are not feasible. Our recent single centre RCT comparing two standardised, concentrated neonatal PN (scNPN) regimens, showed that higher protein and energy intake can improve head growth [8]. We have previously conducted a similar study comparing two iNPN regimens [5]. This offers a unique opportunity to compare the efficiency of iNPN and scNPN regimens using RCT data. We hypothesised that the two scNPN regimens would achieve a higher percentage of the target parenteral protein intake than their corresponding iNPN regimens.

## 2. Methods

This analysis used data obtained during two previously ethically approved and published RCTs that shared the same single site, similar eligibility criteria and other methodology [5,8]. The key difference in methodology was in aqueous PN administration. The first study [5] (2004–2006) compared standard and high target parenteral protein intakes in an iNPN regimen (Table 1). The policy was to introduce PN within 24 h (where possible) with a starting dose of 1 g/kg/day increasing by 1 g/kg/day every 48 h until the target dose was achieved. The iNPN was individually prescribed (non-blinded) and manufactured each day for each patient after the fluid requirements had been clinically assessed and laboratory biochemistry were obtained. The nutrient content of this bespoke aqueous PN was determined by randomization group and designed to meet the total fluid and nutrient content described in Table 1 where possible. Prescribing lower volumes while maintaining nutrient content was permissible but only in accordance with existing stability/safety data for the prescribed electrolyte content. The second study [8] (2009–2012), compared standard and high target parenteral protein intakes using an scNPN regimen (Table 1). The policy was to introduce PN within 6 h of birth with a starting dose of 1.8 g/kg/day increasing daily to achieve the target dose by the end of day 4. The scNPN was prescribed from one of 3 premanufactured standard bags with standard concentrated nutrient content allocated by randomization (prescriber blinded). Variations in fluid balance were managed using a supplementary glucose infusion where possible. Electrolyte management is described below. Both studies increased the energy intake in the high-protein groups (using increased glucose and lipid) to maintain the protein:energy ratio.

The policies were used to calculate a daily target PN protein intake for each infant according to their original treatment allocation. The earlier and faster introduction of scNPN compared with iNPN resulted in much higher target protein intakes for the scNPN groups in the first 5 days of life. Daily parenteral and enteral protein intake data were used to calculate actual daily parenteral and total protein intake for the first 15 days of life. After day 15, the majority of infants in both RCTs were not PN dependent. Both studies shared the same intravenous fluid guidelines (Table 1) and aimed to introduce enteral feeds (maternal breast milk if available) as soon as possible. Intravenous fluid requirements were not reduced until enteral intake exceeded 15 mL/kg/day. In the scNPN RCT, the supplementary glucose infusion was weaned first and then, the aqNPN volume. In the iNPN RCT, reducing the aqNPN volume was the only option. Both policies had the same biochemical monitoring protocols, plasma electrolyte targets and approach to hyperglycaemia (insulin therapy) and hyperlipidaemia. Electrolyte/mineral deficiency was managed through individualised prescribing and/or bespoke supplementation in the iNPN RCT. In the scNPN RCT, electrolytes were managed by first selecting the most appropriate standard aqueous bag (bag 1: no electrolytes; bag 2: maintenance electrolytes; bag 3: bag 2 with additional sodium) and then using standardised electrolyte infusion supplements (in 10% glucose) if required. These were administered alongside the scNPN using standard compatibility data. The numbers of days on which each infant received supplementary electrolyte infusions was recorded, as was the actual electrolyte intake (mmol/kg/d).

### Analytical Methods

The collection of growth data is described in the original RCT publications [5,8]. Change in weight between birth and 36 weeks corrected gestational age (Δweight 36wkCGA) and equivalent change in standard deviation score (Δweight SDS 36wkCGA) shared the same methodological approach in both studies. Similarly, change in occipitofrontal circumference (OFC) over the same period is presented as ΔOFC 36wkCGA and ΔOFC SDS 36wkCGA.

To calculate target attainment, the days where maximum aqNPN intake was required (defined by protocol and enteral intake) were identified for each infant. The actual daily parenteral protein intake for all maximum aqNPN days was divided by the target parenteral protein intake for all maximum aqNPN days to obtain percentage target attainment for the study period (day 1–15). This ensured each infant contributed a single data point to their group. The combination of incremental introduction of parenteral and enteral protein means that target attainment may be influenced by postnatal age. To explore this, the study period was divided into three phases:Initiation and randomisation phase (day 1–5).Maximum PN phase (day 6–10).Transition to mainly enteral feeds phase (day 11–15).

Target attainment (%) was calculated for the phases as above. Infants with no days of maximum aqNPN-dependence during a particular phase were excluded from analysis for that phase.

The target attainment calculations were then repeated for all PN days where enteral feeds were less than 75 mL/kg/day rather than PN days where aqNPN was maximal. This provides information about the early transition phase from NPN to enteral feeding. The target protein intake was kept as the maximum parenteral protein intake, but the actual protein intake included both parenteral and enteral sources.

Control iNPN and scNPN groups and intervention iNPN and scNPN groups were compared using an unpaired *t*-test for normally distributed dated and Mann–Whitney U tests for non-parametric data (percentage target attainment). Other comparisons used Fisher’s exact test as appropriate.

## 3. Results

The demographic data for the two RCT have been reported in detail previously [5,8]. There were no differences at randomisation in birthweight iNPN and scNPN regimens when comparing the two control groups or two intervention groups (Table 2). However, gestational ages were slightly lower in the iNPN groups. All growth outcomes were better in the scNPN groups when compared to their corresponding iNPN groups (Table 2). However, these differences were only statistically significant for weight gain (*p* = 0.050; control groups only) and head growth (*p* < 0.001; intervention groups only).

All 14-day survivors contributed to day 1–15 and day 1–5 data in the scNPN group but only data from 36 weeks CGA survivors was available in iNPN groups. The target attainment for parenteral protein intake in infants receiving full PN (Table 3) was statistically significantly higher with scNPN versus iNPN regimens for both standard (control) and high (intervention) protein intakes. The effect was greatest during day 1–5 despite higher target parenteral protein intakes set by the scNPN protocol.

The target attainment for parenteral protein intake in infants still dependent on NPN for the majority of their nutritional needs (Table 4) was statistically significantly higher with scNPN versus iNPN regimens for both standard (control) and high (intervention) protein intakes. Target attainment exceeds 100% in many infants receiving scNPN. This is partly because, as enteral feeds are introduced, the supplementary glucose infusion is reduced first, meaning that parenteral protein intake is preserved at maximum intake despite enteral protein being introduced.

Standardised supplementary electrolyte infusion usage for scNPN groups is shown in Table 5. There were 1040 and 1072 NPN days in the intervention and control groups, respectively. It shows that potassium and phosphate infusion usage is higher in the intervention group. More detailed evaluation of the timing of this phenomenon shows that the greatest difference occurs during the 6–10-day postnatal epoch. This coincides with maximal parenteral amino acid intake (Table 3).

## 4. Discussion

This is the first paper to describe the effect of different methods of PN administration on the extent to which targets for parenteral protein (amino acid) administration are attained. It is clear that the scNPN regimen achieves parenteral protein intakes much closer to the intended parenteral protein target intake than the iNPN regimen, even though those target intakes are higher. A median target attainment of 75% (in the iNPN group) indicates that half the infants are “losing” >25% prescribed parenteral protein intake when receiving maximum NPN. Not only does this contribute to suboptimal nutrition but it introduces unintended and unpredictable variation in nutritional intake between patients. The principle of concentrating the standardised NPN formulation to prevent this and optimise nutrient intake has been demonstrated. There is limited evidence that growth outcomes may be improved as a result. The reasons for lower target attainment with iNPN regimens are multifactorial but include limited aqNPN availability and the effect of additional drug infusions and fluid restriction. The iNPN regimen did not use computer-aided prescribing [21] or other decision support systems [42] which may have ameliorated the nutritional deficits with iNPN. Management of PN intolerance (e.g., hyperglycaemia, hyperlipidaemia) can also affect nutritional intake but the protocol for managing these challenges were the same in both RCT. We have previously [39] shown that immediate introduction of AA after birth (scNPN protocol) is associated with less insulin use than protocols that allow up to 24 h before starting AA (iNPN protocol). Our insulin treatment protocol allowed the scNPN formulations to be designed with a single concentration glucose for each group. However, management of hyperglycaemia varies greatly across neonatal services, as shown in UK surveys [43]

This paper illustrates the strengths of using the principle of target attainment focused on comparing expected versus achieved parenteral nutrient intake. It has the potential to raise national standards for NPN provision by including nutritional targets (e.g., >75% preterm infants must achieve >90% target parenteral protein intake) within neonatal service specifications to drive regular audits of NPN nutritional intakes. Setting regional and national standards are important initial steps to quality improvement [44]. This has the advantage of avoiding rigid standards for the amount of parenteral nutrients delivered (e.g., 3 g/kg/day or 4 g/kg/day parenteral protein), where the evidence base is still incomplete and focuses instead on ensuring whatever local nutritional standard is set, that this is consistently delivered to every infant. This addresses a key failure identified in national audits of NPN [15,16,17,18]. Unintended variation not only compromises nutrition but increases wastage and is an important safety issue (see below). The approach to identifying epochs that define days by dependence on NPN rather than enteral does have limitations. Substantial numbers of patients are excluded from the later epochs reducing statistical power and increasing the risk of bias. The calculation for parenteral protein target attainment during the first half of transition to enteral feeds (Table 4) helps to address this problem by reducing the number of excluded infants. The overall approach does avoid distorting the parenteral attainment targets with any potential failures in the enteral feeding regimen (including the transition period). This allows the source of unintended variation in nutrient intake to be clearly identified.

Suboptimal target attainment has important implications for patient safety as the efficiency of protein intake is a surrogate measure of all other aqNPN components, including trace elements and maintenance electrolytes and minerals. Thus, electrolyte derangement in the very preterm infant is more likely if half of infants are receiving >25% less maintenance electrolyte than intended as described in this paper for the iNPN regimen. The subsequent unpredictable variation in target electrolyte intake (and so risk of plasma electrolyte derangement) has potential implications for patient safety. The use of standardised supplementary electrolyte/mineral infusions as part of the scNPN regimens allows rapid response to electrolyte/mineral deficiencies without compromising nutrition. The standardised supplementary electrolyte/mineral infusion usage data for the scNPN provides important information about potential additional workload and costs for this approach to correcting electrolyte deficiency. Standardising allows infusions to be premanufactured reducing costs, workload and reducing risks. These data also provide the information to optimise the future scNPN electrolyte/mineral content to meet the needs of the maximum number of infants and so minimise the need for supplementary mineral/electrolyte infusions. Hypokalaemia and hypophosphataemia are associated with increased parenteral AA provision [45,46,47,48]. Delayed introduction of phosphate (as in the scNPN regimen protocol) has been shown to lead to hypophosphataemia [49]. Quantifying additional electrolyte/mineral needs following the introduction of standardised NPN regimens helps to optimise future formulations [50]. We have subsequently modified the scNPN regimen to introduce potassium and phosphate sooner and increased the maintenance dose of phosphate.

One of the strengths of this study are that the data were collected as part of two randomised controlled trials in the same centre, using similar, clearly defined fluid, parenteral and enteral nutrition regimens. The two standard protein/energy groups and two high protein/energy groups had very similar prescribed intakes as part of the NPN guideline and the management of PN intolerance (insulin treatment for hyperglycaemia and triglyceride monitoring) was the same. Nevertheless, the compared groups were in different RCT from different epochs (5 years apart) raising the possibility of confounding factors arising from other aspects of clinical management. In addition, the scNPN protocol has a more aggressive approach to introducing and increasing AA than the iNPN regimen. While this has no effect on target attainment, it has clear potential to affect growth outcomes [51,52], greatly limiting their interpretation. We did not assess target attainment for energy because the primary purpose of this paper was to identify unintended variation. The presence of protein in only one fluid compartment (the aqueous NPN bag) in both regimens is ideally suited to this purpose. The much larger intended variation in glucose intake (due to the clinical need to adjust fluid intakes and use drug infusions in 10% glucose) makes interpreting unintended variation virtually impossible. Both RCT used identical, separate lipid infusions in their regimens so there were no regimen differences to compare. Previous studies comparing iNPN and standardised NPN have mainly been before and after a change in practice. Those that have indicated that standardised NPN regimens are nutritionally inferior have usually used standardised formulations designed to achieve low protein and/or energy intakes [20,27], particularly if the standardised formulations are restricted to those commercially available [53]. This makes interpreting the outcomes difficult because the nutrient intake targets are different in the iNPN and standard NPN groups. Our study indicates there is no reason to limit the protein/energy intake in the standard formulation, particularly given the option to concentrate the formulation. Since the scNPN RCT completed, “all in one” NPN bags have become commercially available [54]. In theory, the principles of the scNPN regimen could also apply to these formulations; however, adding intravenous lipid to concentrated aqNPN formulations may affect both the flexibility and stability of such a scNPN regimen.

The findings in this paper are consistent with several studies that have shown benefits for standardised PN formulation [27,28,29,30,31,32,33] and improve macronutrient intake when compared to iNPN regimens [34,35]. These studies also have the benefits of standardisation of aqPN, including with concentration, on total protein intake and cost. However, these studies have not provided the detailed efficiency and safety data described in this paper, which is required to inform those procuring standardised PN formulations for neonatal services.

## 5. Conclusions

ScNPN regimens improve parenteral protein intake and have the potential to improve patient safety when compared to iNPN regimens by increasing macronutrient target attainment and reducing variability during aqNPN administration.

## Figures and Tables

**Table 1 nutrients-11-02167-t001:** Description of parenteral fluid components in each randomised controlled trial (RCT) group demonstrating the concentration of parenteral protein with standardised concentrated neonatal parenteral nutrition (scNPN) [8] compared with individualized NPN (iNPN) [5]. The total daily fluid intake policy was identical in both RCT with clinicians restricting or increasing fluids based on individual patient need.

	Control Groups		Intervention Groups	
	iNPN	scNPN	iNPN	scNPN
Maximum aqPN protein content (g)	3.0	2.8	4.0	3.8
Maximum aqPN volume (mL)	135	85	135	100
Supplementary fluid volume (mL)	0	50	0	30
Intravenous lipid volume (mL)	15	15	20	20
Fluid regimes for infants <1 kg (>1 kg)				
Day1 (mL/kg/day)	90 (60)	90 (60)	90 (60)	90 (60)
Day 2 (mL/kg/day)	120 (75)	120 (75)	120 (75)	120 (75)
Day 3 (mL/kg/day)	150 (90)	150 (90)	150 (90)	150 (90)
Day 4 (mL/kg/day)	150 (120)	150 (120)	150 (120)	150 (120)
Day 5 (mL/kg/day)	150 (150)	150 (150)	150 (150)	150 (150)

**Table 2 nutrients-11-02167-t002:** Growth data comparing change in weight (Δweight; g or standard deviation score; SDS) and change in head circumference (ΔOFC; mm or standard deviation score; SDS) between birth and 36 weeks corrected gestational age (36wkCGA) in control iNPN and scNPN groups and intervention iNPN and scNPN groups.

**Control Groups**	**iNPN**	**scNPN**	***p*-Value**
Infants randomised	74	76	
Birthweight	914 (219)	884 (183)	0.36
Gestation	26.2 (1.5)	26.8 (1.5)	0.016
Survivors at 36wkCGA	65	64	
Δweight (g) 36wkCGA	1011 (264)	1080 (306)	0.18
Δweight (SDS) 36wkCGA	−1.02 (0.70)	−0.74 (0.90)	0.050
ΔOFC (mm) 36wkCGA	69 (13)	71 (14)	0.41
ΔOFC (SDS) 36wkCGA	−0.03 (0.94)	+0.18 (0.97)	0.21
**Intervention Groups**	**iNPN**	**scNPN**	***p*-Value**
Infants randomised	68	74	
Birthweight	911 (224)	900 (158)	0.73
Gestation	26.0 (1.5)	26.6 (1.4)	0.015
Survivors at 36wkCGA	59	64	
Δweight (g) 36wkCGA	1170 (311)	1160 (268)	0.85
Δweight (SDS) 36wkCGA	−1.00 (0.84)	−0.86 (1.03)	0.41
ΔOFC (mm) 36wkCGA	64 (14)	77 (13)	<0.001
ΔOFC (SDS) 36wkCGA	−0.46 (1.47)	+0.64 (0.83)	<0.001

**Table 3 nutrients-11-02167-t003:** The target attainment for protein intake compared between the iNPN and scNPN RCTs. Target attainment (%) defined as median (IQR) percentage of target protein intake achieved.

Parenteral Protein Intake (g/kg/day) in Infants Targeted to Receive Maximum aqNPN				
	Control Groups		Intervention Groups	
	iNPN	scNPN	iNPN	scNPN
Day 1–15				
Number of infants	65	73	59	68
Mean (sd) protein intake	1.31 (0.52)	2.36 (0.18)	1.61 (0.70)	2.90 (0.38)
% target intake	77 (67–85)	94 (91–96)	75 (66–85)	94 (87–97)
Day 1–5				
Number of infants	65	73	59	68
Mean (sd) protein intake	0.74 (0.26)	2.05 (0.23)	0.79 (0.26)	2.29 (0.36)
% target intake	62 (53–75)	91 (86–94)	68 (56–77)	91 (84–96)
Day 6–10				
Number of infants	48	67	45	62
Mean (sd) protein intake	2.49 (0.36)	2.74 (0.10)	2.74 (0.44)	3.64 (0.35)
% target intake	87 (75–92)	99 (97–100)	80 (67–87)	98 (95–100)
Day 11–15				
Number of infants	20	39	28	38
Mean (sd) protein intake	2.63 (0.59)	2.69 (0.18)	3.46 (0.51)	3.56 (0.44)
% target intake	93 (89–98) *	97 (94–100) *	86 (83–92)	98 (95–99)

Comparison made at two protein doses: standard (control) and high (intervention) in infants on no/minimal enteral feeding (full PN). For all iNPN versus scNPN comparisons, *p* < 0.01 except the pair marked with an asterisk (*), where *p* < 0.05.

**Table 4 nutrients-11-02167-t004:** The target attainment for protein intake compared between the iNPN and scNPN RCTs. Target attainment (%) defined as median (IQR) percentage of target protein intake achieved. Comparison made at two protein doses: standard (control) and high (intervention) in infants receiving <50% enteral feeds (<75 mL/kg/d). For all iNPN versus scNPN comparisons, *p* < 0.01.

Total Protein Intake (g/kg/day) in Infants Receiving <75 mL/kg/day Enteral Nutrition (PN-Dependent)				
	Control Groups		Intervention Groups	
	iNPN	scNPN	iNPN	scNPN
Day 1–15				
Number of infants	64	73	60	68
Mean (sd) protein intake	1.76 (0.37)	2.63 (0.19)	2.11 (0.46)	3.19 (0.30)
% target intake	85 (77–91)	102 (98–106)	81 (73–88)	99 (92–101)
Day 1–5				
Number of infants	64	73	60	68
Mean (sd) protein intake	0.82 (0.25)	2.11 (0.24)	0.88 (0.27)	2.36 (0.36)
% target intake	73 (57–82)	93 (87–97)	74 (63–82)	93 (87–98)
Day 6–10				
Number of infants	64	73	60	68
Mean (sd) protein intake	2.60 (0.36)	3.00 (0.24)	2.86 (0.47)	3.83 (0.39)
% target intake	91 (82–97)	107 (102–114)	84 (77–90)	103 (99–105)
Day 11–15				
Number of infants	30	58	44	58
PN protein intake	2.68 (0.50)	3.02 (0.28)	3.18 (0.74)	3.63 (0.50)
% target intake	95 (90–102)	109 (101–116)	85 (81–93)	99 (90–104)

**Table 5 nutrients-11-02167-t005:** Comparison of standardised supplementary electrolyte infusion usage in control and intervention scNPN groups.

scNPN	Control	Intervention	*p*-Value
Number (%) of supplementary infusion days			
Sodium	159 (15)	137 (13)	
Potassium	38 (4)	94 (9)	
Phosphate	87 (8)	158 (15)	
Calcium	15 (1)	22 (2)	
Magnesium	2 (0.2)	15 (1)	
Number (%) infants with supplementary infusions			
Sodium	37 (49)	40 (54)	0.51
Potassium	18 (24)	38 (51)	<0.0001
Phosphate	26 (34)	55 (74)	0.0007
Calcium	5 (7)	8 (11)	0.40
Magnesium	1 (1)	8 (11)	0.02
Number (%) infants with supplementary potassium			
Day 1–5	15	26	0.04
Day 6–10	8	21	0.007
Day 11–15	0	4	0.06
Number (%) infants with supplementary phosphate			
Day 1–5	22	27	0.39
Day 6–10	17	48	<0.0001
Day 11–15	13	7	0.23

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
