# Peer review of "Attainment Targets for Protein Intake Using Standardised, Concentrated and Individualised Neonatal Parenteral Nutrition Regimens"

_nutrients, 2019, doi:10.3390/nu11092167_

Round 1

Reviewer 1 Report

This manuscript presents data based upon two previously published studies of neonatal PN where it was shown that utilizing standardized PN formulations that had increased concentrations of parenteral amino acid and energy improved nutrient intake and appeared to positively impact head growth.

The study shows that using a lower volume more concentrated standardized PN regimen at a predetermined volume of intake while supplementing fluid intake as needed to attain the desired total fluid intake rather than attempting to provide the total fluid intake by way of the the PN solution alone is associated with a better attainment of target protein intakes.

It seems to me that referring to the regimen where PN is meant to provide the total fluid intake as "individualized NPN" is somewhat misleading. Both regimens appear to be standardized regimens as opposed to regimens where the macro-nutrients as well as minerals and electrolytes are ordered on an individual basis daily such as in reference 27 by Smolkin et al where individualized PN was associated with improved growth.

It also does not seem appropriate to compare the results from the two previous studies with regard to growth. Although infants in the study of concentrated standardized PN did have better head growth when comparing the intervention groups between studies the goal for protein intake in the earlier study was significantly lower than in the study of concentrated PN. If the goals were the same there might not have been any difference even though the target goal was achieved at higher rate utilizing the concentrated PN. There are studies that have failed to show that increased AA intake in PN during the first weeks leads to improved growth with differences in AA intake that would appear to  be greater than would have most likely been the case if the goals for AA intake had been the same for the two regimens in this study (Balakrishman et al JPEN, 2018; Clark et al Pediatrics, 2007). There are also some differences in clinical characteristics between infants in the two studies that are not corrected for in the statistical analysis.

There are a few suggestions for  wording:

Page 2, lines 51, 66, and 71: I would change "manufactured" to "compounded"

Page 3, line 96: add "of" before "an iNPN regimen"

Page 3, line 116: change "approach" to "appropriate"

Page 5, Table 3, second to last row 4th column: there is a "9" where it appears there should be  a parenthesis.

Author Response

This manuscript presents data based upon two previously published studies of neonatal PN where it was shown that utilizing standardized PN formulations that had increased concentrations of parenteral amino acid and energy improved nutrient intake and appeared to positively impact head growth.

The study shows that using a lower volume more concentrated standardized PN regimen at a predetermined volume of intake while supplementing fluid intake as needed to attain the desired total fluid intake rather than attempting to provide the total fluid intake by way of the the PN solution alone is associated with a better attainment of target protein intakes.

It seems to me that referring to the regimen where PN is meant to provide the total fluid intake as "individualized NPN" is somewhat misleading. Both regimens appear to be standardized regimens as opposed to regimens where the macro-nutrients as well as minerals and electrolytes are ordered on an individual basis daily such as in reference 27 by Smolkin et al where individualized PN was associated with improved growth.

We agree this could have been clearer. Lines 98-110 added to clarify this is an individualised NPN regimen with daily bespoke manufacture for each individual patient.

It also does not seem appropriate to compare the results from the two previous studies with regard to growth. Although infants in the study of concentrated standardized PN did have better head growth when comparing the intervention groups between studies the goal for protein intake in the earlier study was significantly lower than in the study of concentrated PN. If the goals were the same there might not have been any difference even though the target goal was achieved at higher rate utilizing the concentrated PN. There are studies that have failed to show that increased AA intake in PN during the first weeks leads to improved growth with differences in AA intake that would appear to be greater than would have most likely been the case if the goals for AA intake had been the same for the two regimens in this study (Balakrishman et al JPEN, 2018; Clark et al Pediatrics, 2007). There are also some differences in clinical characteristics between infants in the two studies that are not corrected for in the statistical analysis.

We would agree the growth comparison is limited and flawed. It was not the primary purpose of the paper (not in hypothesis or conclusion) but given there were comparable growth data available at the same time point I think some reviewers would have questioned why it was not in the paper. The protocol difference in early protein intake is an important point. We have now noted this on lines 213 and 269-272 adding more about the limitations of the growth data and the references suggested. Correcting for gestation does not change the comparison of growth outcomes.

There are a few suggestions for wording:

Page 2, lines 51, 66, and 71: I would change "manufactured" to "compounded"

We chose manufacture as it refers to more than just the aqueous bag (traditionally compounded), as the scNPN regimens includes supplementary glucose and electrolyte infusions which are not “compounded”. However, we do not have particularly strong views about this and are happy to be guided by the editor.

Page 3, line 96: add "of" before "an iNPN regimen" corrected

Page 3, line 116: change "approach" to "appropriate" corrected

Page 5, Table 3, second to last row 4th column: there is a "9" where it appears there should be a parenthesis. corrected

Reviewer 2 Report

The authors have made a quite interesting approach to evaluate the appropriateness of two different types of PN in neonates to achieve protein target. The study is based in 2 previously published RCTs from this group. My main consideration is why the y have decided to evaluate only target on protein intake and not considering total energy o different nutrients. It would provide a best idea of the whole picture. In this way, I would  like the authors to provide some additional information:

1, Are there differences in the need of using supplementary electrolyte infusion when comparing standardized vs individualized PN?

2. Are there differences in the rate of metabolic complications (i.e. hyperglycemia, hypere/hyponatremia, etc.) in both types?

3. How much is the burden for nurses in the Unit when using standardized PN regimens?

4. How the authors monitor the stability when mixing standardized mixtures with Dextrose with lytes piggybagging?

5. As concentrate PN solution have a higher Osm, are there higher numbre of complications -ie thomboflebitis, etc?

Author Response

The authors have made a quite interesting approach to evaluate the appropriateness of two different types of PN in neonates to achieve protein target. The study is based in 2 previously published RCTs from this group. My main consideration is why the y have decided to evaluate only target on protein intake and not considering total energy o different nutrients. It would provide a best idea of the whole picture.

While we understand and appreciate the wider interest, the scope of the paper is very specific in using target attainment to assess unintended variation in the iNPN and scNPN regimens. Only protein target attainment allows this because protein is only present in the aqueous NPN bag and no other components of the regimen. The scNPN is designed to reduce unintended variation associated with the aqNPN bag. The energy components are lipid and glucose. Lipid is administered in a separate infusion at the same doses in both studies (and would therefore not offer a mechanism to explore differences between iNPN and scNPN regimens). Glucose administration is subject to large intended variations due to the clinically indicated differences in total fluid requirements and the use of multiple drug infusions, mostly in 10% glucose. Interpreting unintended variation in this context is virtually impossible (and it is extremely complex to define the target glucose intake by the protocol) Thus, while energy intake is of general interest, it does not address the primary interest of the paper. This is now explained in lines 272-278.

In this way, I would like the authors to provide some additional information:

1, Are there differences in the need of using supplementary electrolyte infusion when comparing standardized vs individualized PN?

Standardised supplementary electrolyte infusions were not used in the iNPN regimen, electrolyte requirements were met using individualised daily prescriptions for the aqueous bag (see additions lines 98-110)

Are there differences in the rate of metabolic complications (i.e. hyperglycemia, hypere/hyponatremia, etc.) in both types?

The insulin usage is reported in other publications. We have previously reported the effect of immediate versus slightly delayed AA on insulin usage (ref 39) so any differences in this paper would be difficult to attribute to the difference in the regimen. This has been clarified in the discussion (line 213)

How much is the burden for nurses in the Unit when using standardized PN regimens?

An interesting question but outside the scope of this paper and the original studies.

How the authors monitor the stability when mixing standardized mixtures with Dextrose with lytes piggybagging?

Compatibility data are used as for any drug infusion (added in line 128). This means separate infusions of phosphate where possible (and calcium always administered separately)

As concentrate PN solution have a higher Osm, are there higher numbre of complications -ie thomboflebitis, etc?

The line complication rates for scNPN have been published previously (Whitby 2015).

Whitby T, McGowan P, Turner MA, Morgan C. Concentrated parenteral nutrition solutions and central venous catheter complications in preterm infants Arch Dis Child Fetal Neonatal Ed 2015;100:F250-2.

Round 2

Reviewer 1 Report

The  manuscript has addressed previous suggestions.

Just two word changes to suggest:

Page 3, line 125: change "approach" to "appropriate"

Page 8, line 269: change "not" to "no"

Author Response

Typographical errors corrected.